# Epidemiology of Human Adenovirus in Pakistani Children Hospitalized with Community-Acquired Gastroenteritis under the Age of Five Years

**DOI:** 10.3390/ijerph191912534

**Published:** 2022-10-01

**Authors:** Nazif Ullah Khan, Aamer Ali Shah, Syed Sohail Zahoor Zaidi, Zhi Chen

**Affiliations:** 1State Key Laboratory for Diagnosis and Treatment of Infectious Diseases, Collaborative Centre for Diagnosis & Treatment of Infectious Diseases, The First Affiliated Hospital, National Clinical Research Center for Infectious Diseases, Zhejiang University School of Medicine, 79 Qingchun Road, Hangzhou 310003, China; 2Department of Biochemistry, Faculty of Biological Sciences, Quaid-i-Azam University, Islamabad 45320, Pakistan; 3Khyber Medical College, Khyber Medical University, Peshawar 25120, Pakistan; 4Department of Microbiology, Faculty of Biological Sciences, Quaid-i-Azam University, Islamabad 45320, Pakistan; 5Department of Virology, National Institute of Health, Islamabad 45500, Pakistan

**Keywords:** human adenovirus, prevalence, HEp-2 cells, ELISA, nested PCR

## Abstract

Acute gastroenteritis is the major cause of morbidity and mortality among infants and children around the globe. Along with other enteropathogens, human adenovirus (HadV) is a major etiological agent associated with diarrhea in young children. However, information about the epidemiology of *Adenoviruses* in Pakistan is limited or has not been reported. A total of 1082 stool samples were collected from patients with acute gastroenteritis under the age of five years with symptoms of diarrhea, vomiting, nausea, and abdominal cramps who visited Benazir Bhutto Hospital Rawalpindi and Children’s hospital in Lahore of Punjab Province in Pakistan. Of this, 384 cases with no blood in their stool, negative for Rotavirus, and under the age of five years were recruited in this study. Human *Adenoviruses* were isolated in the human epithelial HEp-2 cell line. Furthermore, adenovirus antigen detection was carried out by an enzyme-linked immunosorbent assay (ELISA), and then all positive and negative samples were confirmed by nested PCR. After inoculating a clear stool supernatant on HEp-2 cell lines, we observed a positive cytopathic effect in 65 (16%) cases. Using an enzyme-linked immunosorbent assay, HAdV antigens were detected in 54 (14.06%) of the clear supernatant from gastroenteritis cases. However, HAdV hexon coding regions were amplified in 57 (14.80%) fecal samples, mainly from patients ≤24 months of age. The findings of this study suggest that adenovirus circulates significantly in the children population under the age of five years and may be the potential etiological factor of acute gastroenteritis in the mentioned cities. This study provides baseline data about the possible role of adenovirus in causing viral diarrhea in children. Further large-scale epidemiological surveys are recommended to better understand disease burden, etiological agents, and its clinical impact across the country.

## 1. Introduction

Gastroenteritis is a self-limiting, watery-diarrheal disease that persists for less than one week with symptoms including vomiting, nausea, loss of appetite, fever, and abdominal cramps. The severity of the disease can lead to dehydration, resulting in hospitalization or even the death of patients [1,2,3].

Diarrheal diseases are a significant public health concern in both developed and developing countries among children under the age of five years [4]. It causes high morbidity and mortality worldwide, with an estimated 1.7 billion cases of gastroenteritis and 0.525 million deaths per year, making it the second leading cause of death among children under the age of five years [1,4,5]. Acute gastroenteritis can affect individuals of any age and presents a significant health risk to those at the extremes of age, including young children and older people [3]. Viral gastroenteritis’s morbidity and mortality rate are higher in developing countries than in developed countries [6].

Diarrheal diseases can be caused by different agents, including parasites, bacteria, viruses, and toxins. It is reported that at least 25 different protozoa and bacteria can cause a similar clinical syndrome, but viral gastroenteritis is more frequent than parasitic and bacterial diarrhea. It is reported that over 75% of cases are reported to be caused by viruses [7,8]. Along with other viruses causing gastroenteritis, human *Adenoviruses* (HadV) are the leading cause of viral diarrhea in children [9,10,11].

Human *Adenoviruses* (HAdV) are naked, medium-sized, icosahedral particles of 70 to 90 nm. Their genome consists of a single linear, double-stranded DNA molecule and belongs to the genus Mastadenovirus of the Adenoviridae family. The genus Mastadenovirus is further divided based on biological, immunologic, and biochemical characteristics into seven species, from A to G. To date, approximately 103 genotypes of *Adenoviruses* have been reported [12]. Among these species, non-enteric genotypes HAdV-3, HAdV-7, HAdV-8, HAdV-31, particularly enteric genotypes, HadV-40, and HadV-41 are reported to be significantly associated with acute gastroenteritis in children younger than five years [13,14]. Because of the unavailability of effective treatments for viral gastroenteritis except for the rotavirus vaccine, regional and local epidemiological data on adenovirus infection are essential for healthcare practitioners and officials to develop and implement appropriate vaccinations and infection control measures [15,16]. Several studies reported different causative viral agents in children with gastroenteritis in Pakistan [17,18]. However, to the best of our knowledge, there has been no investigation on the prevalence of HAdV-related diarrhea and its epidemiology among children hospitalized with community-acquired gastroenteritis in Rawalpindi and Lahore, Punjab Province, Pakistan [19,20]. This cross-sectional study enrolled hospitalized patients with gastroenteritis from two tertiary care hospitals in Rawalpindi and Lahore, Punjab Province, Pakistan. The data generated in this study will be a valuable contribution to the better understanding and management of acute adenovirus gastroenteritis problems in children under the age of five years in Punjab, Pakistan.

## 2. Materials and Methods

### 2.1. Study Design

A one-year descriptive cross-sectional study was designed to assess the periodic prevalence of human adenovirus gastroenteritis in Pakistani children.

During this study, we enrolled children with community-acquired acute gastroenteritis who were five or under five years of age. For the sample collection, we targeted two different tertiary care hospitals, Benazir Bhutto Hospital, Rawalpindi, and Children Hospital Lahore, from two different cities/districts: Rawalpindi (ranked fourth-largest city by population in Pakistan) and Lahore (ranked second-largest city by population in Pakistan) in Punjab province, Pakistan. Acute gastroenteritis is defined as three or more diarrheal episodes with abdominal discomfort for over 24 h. To understand the epidemiological and clinical features of patients, a detailed clinical history of each individual was noted at the time of samples (stool) collection on a specially designed form. In addition, particular attention was given to the presence of features associated with gastroenteritis infection, like diarrhea, vomiting, fever, episodes of diarrhea and vomiting within 24 h, duration of symptoms of gastroenteritis, and dehydration. After collection, each case’s stool samples and clinical records were transported to the Laboratory of Virology and Immunology, National Institute of Health (NIH), Islamabad, for further laboratory testing.

### 2.2. Sample Collection

A total of 1082 stool samples were collected from the two hospitals. Three hundred eighty-four (384) watery-diarrheal samples negative for Rotavirus were selected, and diarrheal samples with blood were excluded from this study. These samples were collected after the patients were admitted to the said hospitals. In addition, stool samples (watery diarrhea) were collected from children with acute gastroenteritis in stool kits. After collection of the stool, the samples were processed, and the stool supernatants were stored at −80 °C [21].

### 2.3. Laboratory Tests for Adenoviruses

During this study, we employed an integrated method [22] of three different techniques which detect infectious enteric viruses accurately and reliably in a short time. Human adenovirus was first detected in stool samples using cell culture and ELISA and then confirmed with nested PCR techniques.

#### 2.3.1. Adenovirus Isolation in Cell Culture

For the isolation of human adenovirus, human laryngeal epithelial carcinoma HEp-2 cell lines were employed. HEp-2 cell lines were cultured in a minimum essential medium containing 5% fetal bovine serum albumin. Before inoculating the supernatant, the growth media was changed to 1 mL of maintenance media containing 2% fetal bovine serum albumin without HEPES buffer. Adenovirus was isolated in HEp-2 cell lines by inoculating 200 µL of clear supernatants under sterile conditions. Moreover, daily observations were made for 3–7 days for the characteristic grape cluster’s cytopathic effect under an inverted microscope.

#### 2.3.2. Enzyme-Linked Immunosorbent Assay

ELISA kit RIDASCREEN^®^ Adenovirus was purchased from R-Biopharm (R-Biopharm AG, Landwehrstr. 54, D-64293, Darmstadt, Germany). Following the manufacturer’s instructions, all samples were screened for detection of human *Adenoviruses*.

#### 2.3.3. Adenovirus DNA Extraction and Nested PCR

For further confirmation of adenovirus infection, HAdV DNA was extracted from all samples using the Qiagen Spin Column DNA extraction kit according to the manufacturer’s instructions and stored at −20 °C.

The extracted HAdV DNA was subjected to a nested polymerase chain reaction, which was completed in two rounds. The primers used in this study were the same as those described previously by Allard et al., 2001 [23].

The first round resulted in 301 bp, while the second round resulted in 171 bp of the amplified products from the Hexon coding region of adenovirus.

The primers used in the first round of amplification, known as outer primers, were hex1deg (5′-GCC SCA RTG GKC WTA CAT GCA CAT C-3′) and hex2deg (5′-CAG CAC SCC ICG RAT GTC AAA-3′). First-round of PCR was carried out in a 50 µL reaction volume, containing 5 µL 10× PCR buffer, 2 µL of 50 mM MgCl_2_, 1 µL of dNTPs, 0.5 µL of each hex1deg and hex2deg primers, 0.2 µL of Taq DNA polymerase, 30 µL template DNA, and a sufficient quantity of DEPC water to make the volume up to 50 µL. In the second round of nested PCR, again, a 50 µL reaction mixture was used, containing all reagents as mentioned for round-1 PCR except the addition of 10 µL of round-1 DNA as a template for second round of nested PCR and the primers known as “nested primers” which were nehex3deg (5′-GCC CGY GCM ACI GAI ACS TAC TTC-3′) and nehex4deg (5′-CCY ACR GCC AGI GTR WAI CGM RCY TTG TA-3′) [23].

The cycling conditions applied for both the PCR steps were 94 °C for 3 min for denaturation, followed by 35 cycles of amplification at 94 °C for 30 s, 55 °C for 30 s, 72 °C for 1 min, and final extension at 72 °C for 5 min and 04 °C for ∞.

#### 2.3.4. Gel Electrophoresis

The amplified PCR product was validated by 2% agarose gel electrophoresis, stained with ethidium bromide, and visualized by using a gel documentation system.

### 2.4. Data Analysis

Statistical analysis of the data was performed by Statistical Package for Social Sciences software SPSS 26.0 (IBM, Chicago, IL, USA). The descriptive statistics were presented as frequency, mean, and percentages. Statistically, a significant difference was assessed by Pearson’s chi-square test, and *p* < 0.05 was considered statistically significant.

## 3. Results

### 3.1. Demographic and Clinical Characteristics of Patients

A total of 384 stool samples negative for rotavirus and norovirus were included in this study. The samples were collected from children with acute watery diarrhea from two tertiary care hospitals in two different cities, Rawalpindi and Lahore, in Punjab province, including Benazir Bhutto Hospital, Rawalpindi, and Children’s Hospital, Lahore. Among these patients, 62% (n = 238/384) were male and 38% (n = 146/384) were female; 73.70% (n = 283/384) of patients were enrolled from Benazir Bhutto Hospital Rawalpindi, and 26.30% (n = 101/384) of patients were from Children’s Hospital Lahore, shown in Table 1.

The median age of all the 384 patients was 19 months (range, 1–51 months). At the time of sample collection, patients were complaining of different clinical symptoms related to gastroenteritis. In total, 100% of the children had watery diarrhea with an average of 5.42 (range 3–7) diarrheal episodes in 24 h, while the mean diarrheal duration was 3.58 days (range 2–6 days). However, 63% of patients had vomiting, with a mean of 4.01 (range 2–8) episodes per 24 h. Likewise, the mean symptoms of vomiting persisted for a mean of 2.91 days (range, 1–7 days). The symptoms of acute gastroenteritis lasted for a mean duration of 4.23 days (range of 3–7 days); see Table 1.

While 16.40% were severely dehydrated, 40.36% were mildly dehydrated, and 12.5% had a fever. Most of the patients recruited in this study (88.94%) were supported by intravenous fluid (IVF), and 11.06% of patients were treated with oral rehydration treatment (ORT) therapy.

### 3.2. Adenovirus Detection

Adenovirus was first detected by cell culture, followed by the ELISA approach, and then confirmed by nested PCR. First, by inoculating the stool supernatants into the HEp-2 cell lines, 16.92% of samples (n = 65/384) showed a positive cytopathic effect. Out of these 65 positive samples, 14.48% (n = 41/283) were from Benazir Bhutto Hospital, Rawalpindi, and 23.76% (n = 24/101) were from Children’s Hospital, Lahore, as we recruited patients from these two hospitals. The difference was statistically significant between the infants from the two cities (χ^2^ = 4.553, *p* = 0.033). Likewise, adenovirus antigen was tested positive in 14.06% of children (n = 54/384). Of these, 14.06% positive infants, 11.30% of patients (n = 32/283) were from Benazir Bhutto Hospital, Rawalpindi, and 21.78% of patients (n = 22/101) were from Children’s Hospital Lahore. The difference in adenovirus infection was still significant between the young patients from the mentioned hospitals of Rawalpindi and Lahore (χ^2^ = 9.752, *p* = 0.008).

A total of 14.85% of children (n = 57/384) were positive for adenovirus on nested PCR, which can be seen in Table 1, including 12.01% of patients (n = 34/283) from Benazir Bhutto Hospital Rawalpindi, and 22.77% of patients (n = 23/101) were from Children’s Hospital Lahore. There was a significant difference in the prevalence of HAdV in hospitals in the two cities, Rawalpindi and Lahore (χ^2^ = 6.815, *p* = 0.009) (Table 2).

#### 3.2.1. Detection According to Gender

Watery diarrheal samples were collected from 62% of males (238/384) and 38% of females (146/384) patients. Among these, a total of 19.74% (n = 47/238) were males, and 6.8% (n = 10/146) were female patients with acute watery diarrhea who tested positive for human adenovirus. There, we observed a significant difference in the positivity rate between the genders for adenovirus (χ^2^ = 11.910, *p* = 0.001). In Benazir Bhutto Hospital in Rawalpindi, 15.3% (n = 28/183) of male and 6% (n = 6/100) of female patients tested positive. The positivity rate was significantly different statistically between male and female patients from Rawalpindi (χ^2^ = 5.291, *p* = 0.021). Similarly, in Children’s Hospital Lahore, 34.54% (n = 19/55) of male and 8.60% (n = 4/46) female patients tested positive for human adenovirus. Consistent with previous results from Rawalpindi, the positivity rate between males and females was significantly different from that of Children’s Hospital Lahore city (χ^2^ = 9.518, *p* = 0.002) (Table 2).

#### 3.2.2. Detection by Age Group

We classified the study population into five age groups in this study. Moreover, the results showed that 78.94% (45/57) of HAdV-infected children were under or equal to 24 months of age. Pearson’s chi-square statistical test showed a significant difference among the five age groups (χ^2^ = 14.177, *p* = 0.007); see Table 2. Among all five age groups, the highest positivity rate for HAdV was noted in children under the age group of 13–18 months (31.5%, n = 12/38), while the second-highest detection rate was observed in children from the age group of 7–12 months (18.94%, n = 18/95). In contrast, the lowest detection rate was observed in children older than 24 months (8.70%, n = 12/137). Similarly, a detection rate of (14.81%, n = 8/54) was observed in the age group of 1–6 months. The second-lowest detection rate was observed in the age group of 19–24 months (11.66%, n = 7/60), which can also be seen in Figure 1.

#### 3.2.3. Seasonality of Adenovirus

Adenovirus diarrheal infections were detected in all months of the year, but the peak positivity rate was observed during the months of September, 40.74% (n = 11/27), January 33.33% (n = 9/27), October 26.08% (n = 6/23), and November 21.42% (n = 6/28), (Figure 2). A total of 21.61% (83/384) patients were studied during winter (December, January and February), 26.30% (101/384) during spring (February, March and April), 31.77% (122/384) during summer (May, June and July), and 20.31% (78/384) during autumn (September, October and November). Statistical analysis shows that there is a great, significant difference in HAdV prevalence season wise (χ^2^ = 22.716, *p* < 0.001). A positive peak rate of 29.48% (n = 23/78) was observed during the autumn, and 19.28% (n = 16/83) of patients were infected during winter. Similarly, 8.91% (n = 9/101) of patients were infected during the spring, and the lowest detection rate was observed at 7.38% (n = 9/122) during summer in this study, which is also shown in Table 2.

## 4. Discussion

Acute gastroenteritis, mainly diarrhea, is one of the most common infections in humans and remains a leading cause of morbidity and mortality in developing and developed countries [11,24,25]. Like other developing countries, Pakistan also faces a high burden of diarrheal infections every single year, and it is reported that about 16% of pediatric deaths are attributed to diarrheal diseases [26]. Pakistan is considered among the top five countries with higher morbidity and mortality due to gastroenteritis [27]. Regardless of this worsening scenario of diarrheal disease, Pakistan’s healthcare system remains underdeveloped, despite receiving assistance from international organizations, such as the World Health Organization (WHO), in combating infectious diseases, including diarrheal infections. More than 20 types of viruses cause gastroenteritis, but pathologically, the most common viruses causing gastroenteritis are group A rotaviruses, *Adenoviruses*, caliciviruses, and astroviruses [10,28]. To the best of our knowledge, this is the first report from Pakistani designed to investigate and understand the role of HAdV infection among diarrheal patients under the age of five in Rawalpindi and Lahore, the two biggest and most populated cities in the province of Punjab, Pakistan.

In the current study, we used an integrated diagnostic approach of three different techniques to isolate and detect human adenoviruses in the study population. First, all stool supernatants were inoculated into HEp-2 cell lines, screened through ELISA, and then confirmed by nested PCR. We observed a positive cytopathic effect in 16% of samples, and HadV antigens were detected in 14.04% of samples. This may be due to the fact that HEp-2 cell lines support the growth of other enteroviruses [29]. At the same time, the HAdV hexon region was amplified in 14.80% of samples through nested PCR. However, nested PCR failed to amplify the hexon region in three of the ELISA positive results samples, which may be due to the false positivity. PCR is more sensitive and specific as compared to ELISA, while ELISA integrated to the nested PCR approach leads to 10 times more sensitive and specific detection of viral pathogens [30]. The results of our study are comparable to other studies from different territories or countries of the world, as the reports indicate that HAdV was responsible for 19.8% of diarrheal patients in Germany [31,32], 15% in the United Kingdom [33], and 15.5% from Saudi Arabia [34]. The prevalence rate in our study was lower than the 23% reported in Northwestern Nigeria [35] and 18.20% in Denmark [36]. Some of the previous studies from neighboring and non-neighboring countries reported a somewhat lower prevalence of HAdV: 4% from Kolkata [37], 7.0% from Aurangabad, and 7.5% from Nagpur, and 9.0% from Delhi, India [38]. In addition, 1.9% was from Dhaka city, Bangladesh [39,40], 5.18% from Iran [41], 5% from France [42], 7.1% from Shanghai, 5% from Xi’an and 4.40% from Lanzhou, China [43,44,45], 7.1% from Korea [46], 8% from Japan [47] and 3.2% from Vietnam [48].

During the current study, we noticed a significant difference in the detection rate of HAdV between the study populations from the two cities in Punjab province. We noted that the prevalence rate was somewhat lower, 12.01%, among the patients from Rawalpindi, compared to 22.77% from Lahore. It may be because Lahore is considered the second biggest city in Pakistan, which is more densely populated and polluted. The sanitation and sewerage system situation in this city is poorer than in Rawalpindi. In the present study, the majority of fecal samples, about 62%, was collected from males and 38% from females.

Contrary to what some investigators have reported, we found a significant gender-based adenovirus infection in general and city wise. Based on our results, we came to know that adenovirus infections were more common in males than females, and our findings were in line with previous studies from Bangladesh and Northwestern Nigeria [35,49,50]. It is reported that due to the physiological difference between males and females, males often mount a weak immune response to viral infection compared to females [51]. As reported by other investigators, we noted in our study that 78.94% of HAdV-infected patients were ≤2 years of age and had the lowest incidence rate in children older than two years [52,53,54,55]. Among age groups, the peak incidence rate of HAdV, 31.50%, was observed in children aged 13–18 months, while 18.94% of the children in the age group of 7–12 months were ranked second for adenovirus infection. Other studies found that children aged 1–6 months are less affected, and these data support the notion that maternal antibodies are effective in protecting children against viral infections in the first six months of life [56].

Seasonal variability is one of the prominent environmental factors that contribute to the epidemiology of infectious diseases, particularly viral disease incidence in human, animal, and plant populations. During this study, we also investigated a possible link between the incidences of HAdV diarrheal infections and the year’s four seasons. This study observed adenovirus infection throughout the year, but a significant seasonal variation was observed. The peak incidence rate was noted in 29.48% of patients during autumn, 19.28% during the winter, and 08.91% during spring. The lowest incidence rate, 7.38%, was observed during the summer season, and our results are in consensus, as reported by other studies [57]. Likewise, we also observed a significant monthly variation of adenovirus infections; a high frequency was noted during September 40.74%, January 33.33%, October 26.08%, and November 21.42%. Our results are in line with already reported data where the HadV infection rate is higher in early autumn. This may be because each virus has a seasonal pattern, with a peak in activity in different seasons and the metrological factors (temperature, humidity, etc.) that favor virus survival and hence the infection rate [58,59,60].

This study, like many others before it, has several limitations. One of the limitations was that we did not include asymptomatic individuals as the control group. In addition, we targeted only two tertiary care hospitals in Punjab province. Hence, the results cannot be generalized to other rural and urban areas and hospitals of the whole province and country. Thirdly, we did not investigate other causes of diarrheal infections, except rotavirus, among the respondents in this study.

## 5. Conclusions

In conclusion, our descriptive cross-sectional study demonstrates that human adenovirus is circulating significantly in the children’s community as a potential etiological factor of acute gastroenteritis with a high disease burden in children under five years. Although this was a small study, the data generated will assist and attract the attention of pediatricians, healthcare professionals, and policymakers to perform systematic and comprehensive large-scale epidemiological surveys in other cities and provinces to understand the actual disease situation in Pakistan, and this will be helpful for timely interventions. In future studies, the inclusion of several surveillance sites, comparison of gastroenteritis patients, and genotyping of adenovirus in Punjab and across the country, both in rural and urban communities, will further assist in understanding the real picture of adenovirus gastroenteritis infections in Pakistan.

## Figures and Tables

**Figure 1 ijerph-19-12534-f001:**
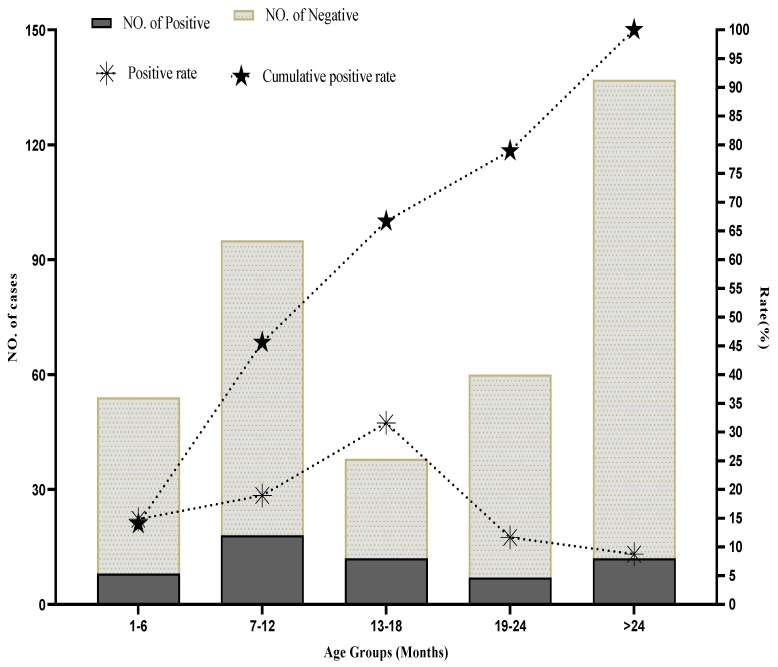
Age group-wise distribution of human adenovirus infection in hospitalized children with gastroenteritis under the age of 5 years.

**Figure 2 ijerph-19-12534-f002:**
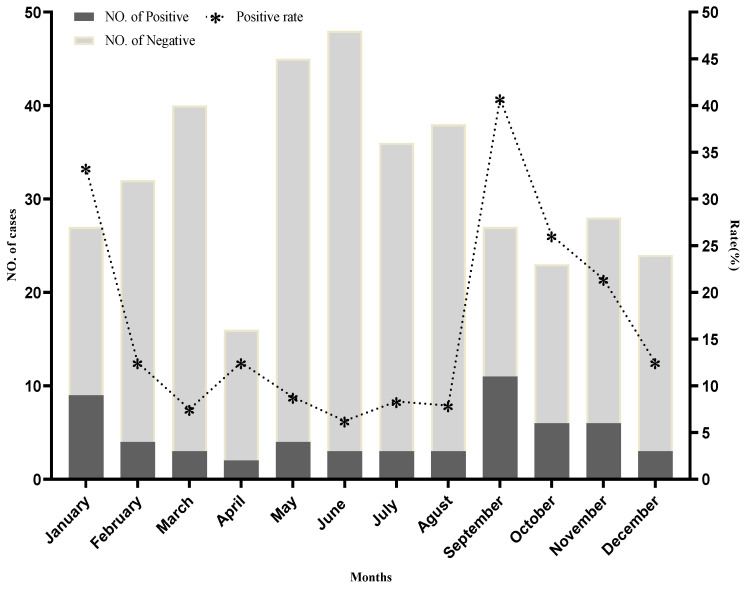
Monthly distribution of human adenovirus infection in hospitalized children under the age of 5 years from Rawalpindi and Lahore Districts, Punjab, Pakistan.

**Table 1 ijerph-19-12534-t001:** Number and percentage of individuals recruited from two hospitals, their demographic and clinical characteristics.

Characteristics	Number of Patients	Percentage	Mean ± SD	Range
Individuals recruited from Benazir Bhutto Hospital, Rawalpindi	283/384	73.70%	NA	NA
Children’s hospital, Lahore	101/384	26.30%	NA	NA
Age (Months)	NA	NA	19	1–51 months
Duration of symptoms (Days)	NA	NA	4.23 ± 1.04	3–7 days
Vomiting episodes per 24 h	NA	NA	4.01 ± 1.72	2–8 days
Vomiting Duration (Days)	NA	NA	2.91 ± 1.59	1–7 days
Diarrhea episodes per 24 h	NA	NA	5.42 ± 1.04	3–7 days
Diarrhea Duration (Days)	NA	NA	3.58 ± 1.2	2–6 days

**Table 2 ijerph-19-12534-t002:** Adenovirus detection by ELISA and Nested PCR in different gender, age groups and seasonality.

Characteristics	Adeno + Ve	Adeno − Ve	Total	*p*-Value
ELISA	54	330	384	-
PCR	57	327	384	
Gender				
Male	47	191	238	0.001
Female	10	136	146	
Age Group				
Group 1 (1–6 months)	8	46	54	0.007
Group 2 (7–12 months)	18	77	95	
Group 3 (13–18 months)	12	26	38	
Group 4 (19–24 months)	7	53	60	
Group 5 (Older than 24 months)	12	125	137	
Seasonality				
Winter	16	67	83	0.001
Spring	9	92	101	
Summer	9	113	122	
Autumn	23	55	78	

## Data Availability

Not applicable.

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
