# Peer review of "Epidemiology of Human Adenovirus in Pakistani Children Hospitalized with Community-Acquired Gastroenteritis under the Age of Five Years"

_ijerph, 2022, doi:10.3390/ijerph191912534_

Round 1

Reviewer 1 Report

                                          REVIEWER COMMENTS

The authors aimed to study the prevalence of Human adenovirus in Pakistani young children with diarrhoea.                                                           Using the culture methods, Elisa assay and molecular techniques, they found respectively 16,92%; 14,06 and 14,85% of overall Human adenovirus prevalence rate. It is a piece of paper that may inform the strategies and policies against Acute gastroenteritis in the region. However more still need to be done to improve the report and reach an acceptable scientific communication.

                             Abstract and text

I recommend an English editing for all the content of this report.

                              Introduction

References 8,9,10 and 11 need to be updated. Please use the recent related publications.

-Lines 74-75:  … Several studies… Give references to that statement.

                          Materials and Methods

-Can the authors motivate the use of three methods for Human adenovirus detection in this study?

-Why did they not work on the asymptomatic patients as controls to assess properly the significant role of Human adenovirus in the study area?

-Lines 140-142: The PCR conditions for the second run of nested PCR should be mentioned.

                                Results

-The tables can be reduced to combine tables 1 and 2, tables 3 and 4.

-Lines 158 and 171: Use while instead of whereas…

-Line 175: Remove rate on the subtitle.

-Numbering of subtopics:

I suggest that 3.3 to be changed to 3.2.1; 3.4 to 3.2.2; 3.5 to 3.2.3.

-Seasonality of Adenovirus:

Line 225: Can the authors explain why there is a peak (40.74%) during the September month?

                               Discussion

-Comment and compare the results of the 3 methods used in terms sensitivity and specificity.

-Lines 251-252: The sentence from …As there… should be removed and rephrase as follows: To the best of our knowledge this is the first report from Pakistan…

-Line 254: Remove …in the light of foregoing… Use therefore

-Line 258: Remove …have…

-Line 275: Correct…During the current study, we noticed

-Line 277: Correct… We noted

-Line 296: 48 is a wrong reference.

-Lines 297-307: Why there is a peak during autumn?                                                                     With note, the discussion is not a merely description of results. The results should be discussed to give the trends and explain the rationale behind.

-Line 309: Correct …we targeted two tertiaries…

                               Conclusion

Line 314: Correct … demonstrate

Line 317: Correct… to attract

Line 318: Correct… the policymakers to perform

Line 320 and 322: … Avoid repetition of word “helpful”.

                                                                                                                                            31/07/2022

Author Response

 Introduction

We updated the references but some relevant studies which have not been published recently and new research study is not available due to which we did not update that reference.

-Lines 74-75:  … Several studies… references provided.

                          Materials and Methods

-Can the authors motivate the use of three methods for Human adenovirus detection in this study?  we mentioned this in the article.

-Why did they not work on the asymptomatic patients as controls to assess properly the significant role of Human adenovirus in the study area?

Yes like most of the study available, we did not include asymptomatic patients as control because our study was a diagnostic kind not a trail. As we aimed to detect HAdVs in patients with Gastroenteritis.

-Lines 140-142: The PCR conditions for the second run of nested PCR should be mentioned. We have mentioned the second round PCR run conditions except the thermal profile as both the steps employ the same thermal profile.

                                Results

-The tables can be reduced to combine tables 1 and 2, tables 3 and 4.

Tables were combines as per suggestions.

-Lines 158 and 171: Use while instead of whereas…

Done

-Line 175: Remove rate on the subtitle.

Done as per suggestions.

-Numbering of subtopics:

I suggest that 3.3 to be changed to 3.2.1; 3.4 to 3.2.2; 3.5 to 3.2.3.

Finished the numbering as per your suggestions.

-Seasonality of Adenovirus:

Line 225: Can the authors explain why there is a peak (40.74%) during the September month? We explain this in discussion section with references.

                               Discussion

-Comment and compare the results of the 3 methods used in terms sensitivity and specificity. Done.

-Lines 251-252: The sentence from …As there… should be removed and rephrase as follows: To the best of our knowledge this is the first report from Pakistan…

Done as you recomended.

-Line 254: Remove …in the light of foregoing… Use therefore

Done

-Line 258: Remove …have…

Done

-Line 275: Correct…During the current study, we noticed

Done as per suggestions.

-Line 277: Correct… We noted

Done

-Line 296: 48 is a wrong reference.

Updated 

-Lines 297-307: Why there is a peak during autumn?                                                                     With note, the discussion is not a merely description of results. The results should be discussed to give the trends and explain the rationale behind. Included a rationale as demanded

-Line 309: Correct …we targeted two tertiaries…

Done

                               Conclusion

Line 314: Correct … demonstrate… Done

Line 317: Correct… to attract…   Done

Line 318: Correct… the policymakers to perform… Done

Line 320 and 322: … Avoid repetition of word “helpful” Done

Reviewer 2 Report

Nazif Ullah Khan and coworkers report findings of a cross-sectional study on the prevalence of human adenovirus (HAdV) in two cohorts of hospitalized children with gastroenteritis in Rawalpindi and Lahore, Punjab Province, Pakistan. The stool samples with no blood, negative for rotavirus and from children under 5 years old from 382 children were used to inspect for cytopathic effect in Hep-2 cell; analyzed by ELISA and nested PCR to detect HAdV hexon protein or DNA sequences, respectively.

The study is of interest because HAdV are relevant gastrointestinal pathogens, and studies of their prevalence and epidemiological impact have not been performed in many countries and geographical regions.

The experimental design is straightforward and the methods and results are presented clearly; however, various aspects, both in terms of the clarity of writing and content can be improved:

1. I recommend rewording the statement in the abstract: "Human adenoviruses were primarily isolated in the human epithelial 25 HEp-2 cell line". Presumably, the HAdV from stool samples were, as a first step, initially cultured and then used for the ELISA and PCR assay; therefore, "primarily" may not be the term of choice.

2. Also in the abstract, I recommend rewording: "The findings of this study revealed that adenovirus is the etiological agent of gastroenteritis in the mentioned cities...". Out of 1082 samples the authors test 382 and find HAdV in 57 through PCR; therefore, HAdVs cannot be said to be "...the etiological agent of gastroenteritis in the mentioned cities..."

3. The description and references cited (5-7, 9, 12-14) on the epidemiology and burden of viral gastroenteritis must be updated.

4. The number of known HAdV genotypes should be updated.

5. I recommend verifying the origin of the Hep-2 cell line. According to ATCC it is no longer considered a laryngeal cell line, but rather derived from HeLa cells.

6. A detailed description of the primers used for the PCR must be included, and the paper by Allard et al included in the References section.

6. The authors should consistently include the potential relevance of gastrointestinal viruses, other than rotaviruses and HAdV, such as astroviruses and caliciviruses, and more importantly describe unambiguously whether the samples were tested for rotaviruses and noroviruses, or only rotaviruses. While in the Abstract, L24, in the Materials and methods, L104, and in the Discussion, L311, testing is only described for rotavirus, in the Results, L153, testing is mentioned for both rotavirus and norovirus.

7. It should be interesting to correlate the samples that produced cytopathic effect with those that were ELISA and PCR positive.

8. I recommend avoiding the unnecessary repetition of the results in the discussion section, rather, in the context of point 7 above, attempt to address the significance of the observations:

a) Did the same samples that were positive for cytopathic effect in the Hep-2 cells result in positive ELISA and PCR tests?

b) Other than the possible explanation of "false-positive" for the three samples that were positive for ELISA and negative for PCR, is it possible that the primers used cannot detect all the HAdV-types? The gastroenteric HAdV species G type 52 was identified after the Allard et al paper.

c) When comparing the results from other countries or regions, L266-268, whenever possible the authors should update the references.

d) Why do the authors describe a worsening scenario of diarrheal disease? Data should be provided to sustain this statement.

e) Reword and clarify L271-274.

Minor points:

Either the term supernate or supernatant (preferably the latter) should be used throughout the manuscript.

L21 should read: vomiting.

L24 should read: five.

L28 reword: After inoculation a clear stool supernates on HEp-2 cell lines,

L44-46 reword: Diarrheal diseases are a major public health concern in both developed and developing countries among children under the age of five years (4). It causes high mortality and morbidity.

L62-64: Adenoviruses should be plural, therefore, should be referred as "their" rather than "it".

L64: Adenoviridae should be in italics.

L78 reword: This study was a cross sectional and enrolled patients...

L85 reword: ...cross-sectional study to assess the period prevalence of human adenovirus gastroenteritis...

L36: The amount in ng, rather than the volume of DNA should be described.

L177-178 reword, as it was the samples not the children that showed positive effect: First, by inoculating the stool supernates into the HEp-2 177 cell lines, 16.92% of children (n = 65/384) showed a positive cytopathic effect.

L309 should read: two.

L316-317 reword or correct: Although this was a small study but the data generated will assist to attracts the attention of pediatricians,...

Author Response

  1. Finished as per recomendation.

  1. Done as per reviewer suggestion.

  1. The description and references cited are updated. 

  1. HAdV genotypes are updated.

  1.  According to ATCC HEp-2 cell lines are thought to be derived from human larynx epidermal carcinoma. But it also contains contamination with HeLa cell lines in the form of Hels marker chromosome and DNA finger printing. But in our article we just mention that HEp-2 cell lines are larynx epidermal carcinoma cell line.

  1. A detailed description of the primers used for the PCR are included, and the paper is cited References section. 

  1. In our research project we just focused on human adenoviruses in samples negative for Rotavirus. Due to limited resources we were unable to focus other virus causing gastroenteritis and I have already mentioned this in study limitations.

  1. All the samples which show positive CPE were not positive on PCR. Nested PCR  also amplified some CPE negative samples.

  1. Recommendation followed as suggested.
  2. a) Yes some samples were amplified successfully which were positive on ELISA and HEP-2 cell culture.
  3. b) The primer used in Allard et al paper are degenerate primers which are designed correspond to the conserved region of HAdV Hexon gene upstream of surface loop. As this region is conserved in all HAdV-subtypes. Therefore, we could say that these primers are capable to amplify all HAdV subtypes with high specificity and sensitivity.  But there is always chances of error and we cannot completely rule out human error etc. 
  4. c)  updated the references have been added.
  5. d) Because Pakistan is among the top 5 countries with highest morbidity and mortality associated with diarrhea. According to CDC diarrhea is the second leading cause of death in Pakistan.
  6. e) Changes has been made.

Minor points:

All the changes have been made according to your valuable suggestions.

Round 2

Reviewer 1 Report

Minor corrections to be addressed before the publication

MATERIALS AND METHODS

- Why did they not work on the asymptomatic patients as controls to assess properly the significant role of Human adenovirus in the study area?

Yes like most of the study available, we did not include asymptomatic patients as control because our study was a diagnostic kind not a trail. As we aimed to detect HAdVs in patients with Gastroenteritis.

Though you were aiming to detect HAdVs in patients in Gastroenteritis, the outcome from such study is to assess to the role of Adenovirus in Acute gastroenteritis. That is why is always better to report on the presence of the virus in the asymptomatic cases as the healthy controls. This should be considered as a limitation of the study.

-Lines 142-146: What you mention there is on the reaction’s mixture (the master mix) not the PCR conditions. Not only the reagents and primers. You need to add them.

RESULTS  

-Table 2: Adenovirus virus detection by Elisa and Nested PCR in different gender, age groups and seasonality

This new table: This is not the way the tables should be combined and set in an epidemiological report. The authors should check in the literature and find the correct set of the variables on one table.

-Line 329-337. The reference 42 cited there is from France. I am not sure that the factors associated with the circulation of enteric virus are similar to those found in Pakistan.

DISCUSSION

-Line 327-328. You did not change the wrong reference as it is still the same title (Walls T et al. 2003).

Author Response

MATERIALS AND METHODS

  • Edited as per suggestion
  • 50ul is a reaction mixture (not a master mix) for each sample in both steps of PCR reactions (First round and nested PCR). And we mentioned all the reagents and PCR thermal profile for both the steps. 

RESULTS  

-Table 2: We combined the two tables into Table-2 by following the design and set of variables from already published literature. Here I will provide you the references which I have a=followed in my report. 

1-doi: 10.2147/CLEP.S246352

2- https://doi.org/10.1093/trstmh/trv042

3-doi:10.1093/trstmh/trv042

But still if the reviewer is not satisfied with this format of table. I would be please to edit it again.   

-Line 329-337. We included a study from Iraq as a reference instead or reference (42). 

DISCUSSION

-Line 327-328. Edited as per suggestion
